# MALDI-TOF mass spectrometry for the identification of freshwater snails from Senegal, including intermediate hosts of schistosomes

**Fatima Zohra Hamlili**[1,2], **Fatou Thiam**[3,4☉], **Maureen Laroche**[1,2☉], **Adama Zan Diarra**[1,2], **Souleymane Doucouré**[3], **Papa Mouhamadou Gaye**[2,3,4], **Cheikh Binetou Fall**[5], **Babacar Faye**[5], **Cheikh Sokhna**[1,2,3], **Doudou Sow**[3,6], **Philippe Parola**[1,2]*

**1** IHU-Méditerranée Infection, Marseille, France, **2** Aix Marseille Univ, IRD, AP-HM, SSA, VITROME, Marseille, France, **3** VITROME, Campus International IRD-UCAD de l'IRD, Dakar, Senegal, **4** Laboratoire de Parasitologie-Helminthologie, Département de Biologie Animale, Faculté des Sciences et Techniques, UCAD, Dakar, Senegal, **5** Service de Parasitologie-Mycologie, Faculté de médecine, Université Cheikh Anta Diop, Dakar, Senegal, **6** Service de Parasitologie-Mycologie, UFR Sciences de la Santé, Université Gaston Berger de Saint Louis, Senegal

☉ These authors contributed equally to this work.
* philippe.parola@univ-amu.fr

**Data Availability Statement:** All relevant data are within the manuscript and its Supporting information files. Snail MALDI-TOF MS database is

## Abstract

Freshwater snails of the genera *Biomphalaria*, *Bulinus*, and *Oncomelania* are intermediate hosts of schistosomes that cause human schistosomiasis, one of the most significant infectious neglected diseases in the world. Identification of freshwater snails is usually based on morphology and potentially DNA-based methods, but these have many drawbacks that hamper their use. MALDI-TOF MS has revolutionised clinical microbiology and has emerged in the medical entomology field. This study aims to evaluate MALDI-TOF MS profiling for the identification of both frozen and ethanol-stored snail species using protein extracts from different body parts. A total of 530 field specimens belonging to nine species (*Biomphalaria pfeifferi*, *Bulinus forskalii*, *Bulinus senegalensis*, *Bulinus truncatus*, *Bulinus globosus*, *Bellamya unicolor*, *Cleopatra bulimoides*, *Lymnaea natalensis*, *Melanoides tuberculata*) and 89 laboratory-reared specimens, including three species (*Bi. pfeifferi*, *Bu. forskalii*, *Bu. truncatus*) were used for this study. For frozen snails, the feet of 127 field and 74 laboratory-reared specimens were used to validate the optimised MALDI-TOF MS protocol. The spectral analysis yielded intra-species reproducibility and inter-species specificity which resulted in the correct identification of all the specimens in blind queries, with log-score values greater than 1.7. In a second step, we demonstrated that MALDI-TOF MS could also be used to identify ethanol-stored snails using proteins extracted from the foot using a specific database including a large number of ethanol preserved specimens. This study shows for the first time that MALDI-TOF MS is a reliable tool for the rapid identification of frozen and ethanol-stored freshwater snails without any malacological expertise.

publicly accessible and can be downloaded with the following DOI number: https://doi.org/10.35088/f605-3922.

**Funding:** This study was supported by the Institut Hospitalo-Universitaire (IHU) Méditerranée Infection, the National Research Agency under the "Investissements d'avenir" programme, reference ANR-10-IAHU-03, the Région Provence Alpes Côte d'Azur and European FEDER PRIMI funding. The field work and maintenance of the snails in our laboratory was carried out with the support of Institute of Research and Development (IRD) through the project "Jeune Equipe Associée à l'IRD (JEAI), ESBILH-SEN". FZH received a grant of PhD scholarship from IHU Méditerranée Infection. The funders had no role in study design, data collection and analysis, decision to publish, or preparation of the manuscript.

**Competing interests:** The authors have declared that no competing interests exist.

## Author summary

Schistosomiasis is a parasitic disease, caused by blood flukes of the genus *Schistosoma*. The infective cercariae are released by freshwater snails belonging to three genera: *Biomphalaria*, *Bulinus* and *Oncomelania*. It is one of the most significant neglected infectious diseases in the world. Snail identification is tremendously important for monitoring snail populations and schistosomiasis. Identification is currently based on morphological criteria and molecular biology, both of which present several drawbacks. Many studies have reported the performance of MALDI-TOF MS, a technology that allows species identification based on their proteins, as a reliable, rapid, and easy-to-use tool in many fields. The aim of our study was to create a snail database and assess the efficiency of MALDI-TOF MS for snail identification. This study shows that MALDI-TOF MS can rapidly identify both frozen and ethanol-stored specimens of different species. These results support the use of MALDI-TOF MS in the context of epidemiological studies in *Schistosoma*-endemic areas.

## Introduction

Human schistosomiasis is a snail-borne parasitic disease caused by blood trematodes of the genus *Schistosoma* [1]. It is considered one of the most significant neglected tropical diseases (NTDs) in the world as it is ranked as the second most endemic parasitic disease in the world after malaria [2]. Almost 800 million people worldwide are at risk of infection and approximately 250 million people are affected, mostly in Africa, including 20 million people who suffer from its severe form [3,4]. The disease has been reported in over 78 countries and is endemic in 52 countries with moderate to high transmission [5,6]. Over 280,000 deaths have been estimated each year [7] and despite efforts that have been made over many years, the number of people affected by schistosomiasis has not significantly decreased [8].

*Schistosoma* species have a complex biological cycle. It includes freshwater snails belonging to the genera *Biomphalaria*, *Bulinus* and *Oncomelania* as intermediate hosts, where the trematode larvae undergo asexual reproduction, and humans as definitive hosts, in which sexual reproduction occurs [9]. The most common disease-causing species are *Schistosoma haematobium*, *S. mansoni*, and *S. japonicum* [6]. Both *S. haematobium* and *S. mansoni* are found in Africa and the Middle East. In the Americas, only *S. mansoni* is found. *Schistosoma japonicum* is found in Asia, more specifically in the Philippines and China [1]. Schistosomiasis transmission is considerably dependent on both the intermediate snail host's spread and the rural development of water resources. Among the intermediate hosts, *Bulinus* species (*Bu. truncatus*, *Bu. globosus*, *Bu. senegalensis*, *Bu. forskalii*, *Bu. camerunensis*, *Bu. africanus* and *Bu. tropicus*) and *Biomphalaria* species (*Bi. pfeifferi*, *Bi. choanomphala*, *Bi. alexandrina*, *Bi. sudanica*) have been reported to be the main intermediate hosts of *S. haematobium* and *S. mansoni* respectively in Africa [10]. *Biomphalaria pfeifferi* is the only species that transmits *S. mansoni*, while *Bu. umbilicatus*, *Bu. globosus*, *Bu. senegalensis* and *Bu. truncatus* are involved in *S. haematobium* transmission [11,12]. In Senegal, two species of schistosomes are found: *S. mansoni*, causing intestinal schistosomiasis, and *S. haematobium*, known as the main cause of urinary schistosomiasis [13].

The need for precise identification of intermediate snail hosts is crucial for accurate epidemiological studies and the control of schistosomiasis, and also facilitates the development of new effective control strategies [14].

Snail classification relies mainly on morphological criteria. However, correct identification of *Bulinus* snails to the species level based on morphology alone can be challenging [15,16]. It also requires tremendous expertise, as well as undamaged specimens and specific documentation. In addition to morphological methods for identifying snail species, molecular methods such as sequencing the snail's Internal Transcribed Spacer 2 region (*ITS2*) and cytochrome c oxidase subunit I gene (*COI*) have also been used [17,18]. Molecular approaches are, however, limited by being time-consuming and by their relatively high running costs. The lack of reference sequences in GenBank may also hamper their use.

Over the course of the last decade, MALDI-TOF MS (Matrix-Assisted Laser Desorption/Ionization Time-of-Flight Mass Spectrometry) has revolutionised clinical microbiology by allowing the rapid and accurate classification of many microorganisms (parasite, fungi and bacteria) [19–26]. More recently, MALDI-TOF MS has emerged as a reliable research tool in entomology, as it has been successfully applied to several hematophagous arthropods using specific body parts [27–29]. To the best of our knowledge, this innovative tool has been firstly used for the identification of scallops [30], but it has not been yet applied to identify snails of medical importance. The aim of this study, therefore, was to assess the efficiency of this proteomic tool for the identification of frozen and ethanol-stored snails' specimens, including species involved in schistosomes transmission.

## Methods

### Ethics statement

The study has received approval from the Comité National d'Ethique du Sénégal pour la Recherche en Santé (CNERS), number: 000017/MSAS/DPRS/CNERS. Date: 2018-03-15.

### Snails

Snails were collected in schistosomiasis endemic areas in Senegal. The study was conducted from 2018 to 2019 in Niakhar and Richard Toll (Fig 1). Richard Toll is a town lying on the southern bank of the Senegal river, located between latitude 16˚27 north and longitude 15˚42 west. The average daily temperature is 29˚C, varying on average between a minimum of 22˚C and a maximum of 36˚C. The rainiest months are July, August and September. Niakhar (14˚30 N, 16˚30 W) is a rural area, located in central Senegal, 135 km east of the capital Dakar. It is situated in a Sahelian-Sudanese climatic region, with temperatures ranging from 24˚C in December to 30˚C in June [31].

In Richard Toll, snail sampling was carried out in the Taouey channel of the Senegal river while in Niakhar, snails were collected from temporary ponds. Snails were collected from aquatic plants, dead leaves and branches, and any solid objects in the water. Recovered snails were then transported to the laboratory in pre-labelled plastic containers which identified the collection point (name of the locality, date of collection). They were then rinsed to remove the mud and were counted. The taxonomic status of the sampled individuals was assessed by microscopy (Zeiss Axio Zoom.V16, Zeiss, Marly-le-Roi, France) at the species level using the identification keys based on snail shell morphology [32,33]. Each identified individual snail was then placed in a glass tube containing 10 ml of filtered water and was exposed to electric light for between 30 and 40 minutes as part of a *Schistosoma* cercarial shedding test [34,35]. Negative individuals were reared in the laboratory until adult offspring were obtained, some of which were then kept at -20˚C (laboratory-reared snails) for proteomic and molecular studies. Besides, the second sample of snails collected from the field and stored at -20˚C or in 70% v/v ethanol was also used. Other frozen laboratory-reared snail specimens of the first generation were also used for the study. Snail specimens were transported carefully to ensure that they

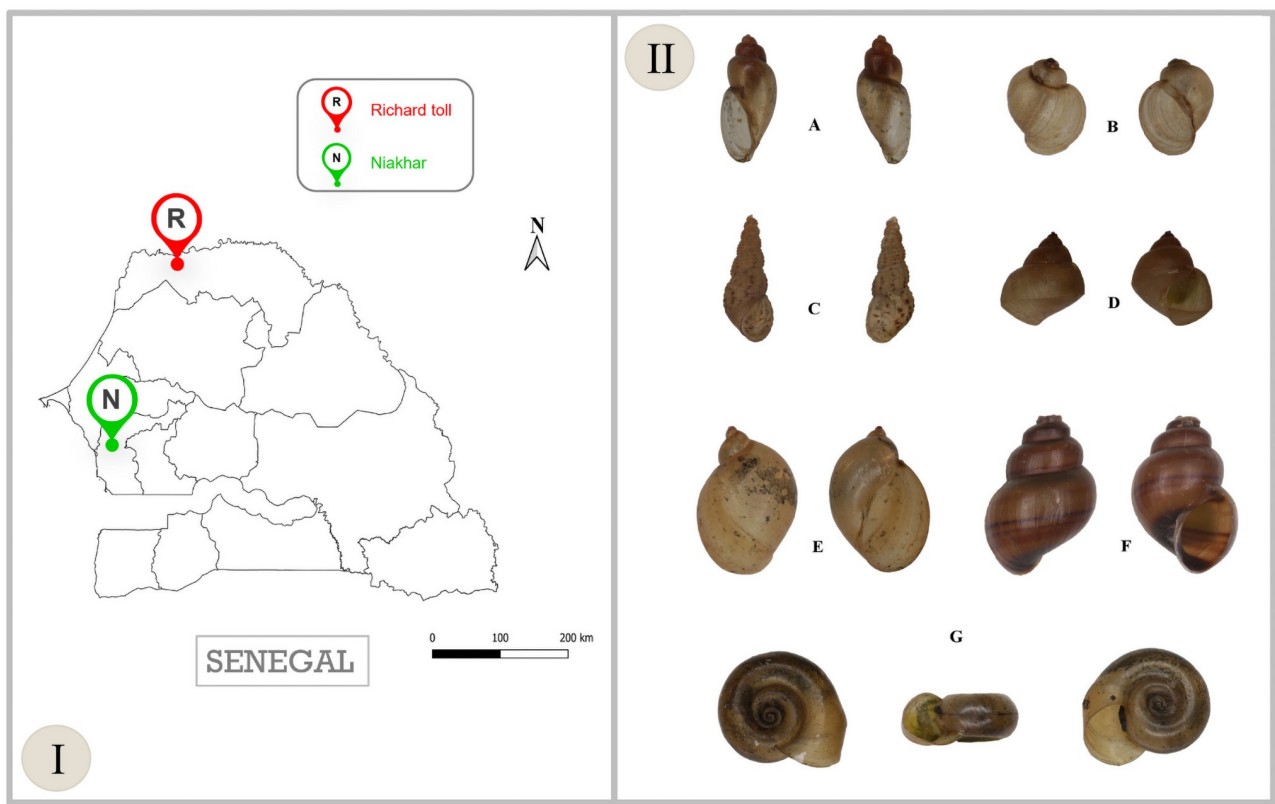

**Fig 1. Map of Senegal showing the snail collection sites and the list of freshwater snails collected from two sites in Senegal.** (I) The map of Senegal showing the sites from which the sample collections were conducted. "R" refers to species collected in Richard Toll; *Biomphalaria pfeifferi* (239), *Bulinus truncatus* (127), *Lymnaea natalensis* (38), *Melanoides tuberculata* (37), *Bellamya unicolor* (20), *Cleopatra bulimoides* (1). "N" refers *to Bulinus senegalensis* (68) collected in Niakhar and morphologically identified. (II) shows the shells of freshwater snails confirmed molecularly: *Bu. forskalii* (A), *Bu. truncatus* (B), *M. tuberculata* (C), *Be. unicolor* (D), *L. natalensis* (E), *C. bulimoides* (F) and *Bi. pfeifferi* (G). The map used in this study are not subject to copyright. Map of Senegal: QGIS.org, 2021. QGIS Geographic Information System. QGIS Association. http://www.qgis.org.

remained intact on their way to our laboratory in Marseille (France), for proteomic and molecular studies.

## MALDI-TOF MS protocol optimisation

**Frozen snail specimens.** Of the hundred specimens preserved at -20˚C, *Bi. pfeifferi* (n = 15), *Bu. truncatus* (n = 13) and *Bu. senegalensis* (n = 3) were used for MALDI-TOF MS protocol development, which included five tested protocols. Snails' shells were carefully removed, and the snails were dissected under a Leica ES2 Stereo Microscope 10x/30x using a new sterile blade to retrieve the foot and the head. The snail's foot was selected for MALDI-TOF MS protocol optimisation because it was stated to be a good source of proteins than other tissues [36]. The dissected parts were rinsed successively with 70% ethanol followed by distilled water for 2 minutes and were dried on sterile filter paper for MALDI-TOF MS analysis. The remaining parts were conserved at -20˚C for molecular biology and supplementary analysis.

We tested five different protocols using only two body parts (the head and/or the foot), homogenized in an extraction solution composed of a mix (50/50) of 70% (v/v) formic acid (Sigma-Aldrich, Lyon, France) and 50% (v/v) acetonitrile (Fluka, Buchs, Switzerland), according to the standardized automated settings previously described [37–39]. The difference between these protocols is not only in terms of the body parts used but also in the extraction

**Table 1. The list of protocols and optimized parameters for all sample storage conditions.**

| Frozen specimens | Species (morphological ID) | Compartment | Modes of sample homogenisation |
|---|---|---|---|
| Protocol H1 | *Bu. truncatus*<br>*Bi. pfeifferi* | Head | Mix (40µl)-tungsten beads |
| Protocol H2 | *Bu. senegalensis*<br>*Bi. pfeifferi* | Head | Mix (40µl)-glass beads |
| Protocol H3 | *Bu. truncatus*<br>*Bi. pfeifferi* | Head | Mix (30µl)-glass beads |
| Protocol FH | *Bi. pfeifferi* | Foot-Head | Mix (30µl)-glass beads |
| Protocol F | *Bu. truncatus*<br>*Bi. pfeifferi*<br>*Bu. senegalensis*<br>*Be. unicolor*<br>*C. bulimoides*<br>*M. tuberculata*<br>*L. natalensis* | Foot | Mix (30µl)-glass beads |
| **Ethanol-stored specimens** | **Species** | **Compartment** | **Modes of sample homogenisation** |
| Protocol F1 | *Bi. pfeifferi* | Foot | Mix (30µl)-glass beads |
| Protocol F2 | *Bi. pfeifferi* | | Mix (30µl)-tungsten beads |
| Protocol F3 | *Bi. pfeifferi*<br>*Bu. truncatus*<br>*Bu. senegalensis* | | Mix (15µl)-glass beads |

solution volume (30 μl or 40 μl) and the homogenisation method: glass beads ≤ 106μm (Sigma Aldrich, Saint-Louis, Missouri, USA) or tungsten beads (Qiagen GmbH, Hilden, Germany) (Table 1). All preparations were homogenised using a Tissue Lyser II device (Qiagen, Germany) with optimised parameters (three 1-minute cycles at a frequency of 30Hertz) [37]. The protocols were assessed initially on the basis of intraspecies reproducibility, spectra intensity ≥ 3.000 arbitrary units [a.u.], the number of MS peaks and the simplicity of the protocol. The different protocols tested had several variables including the volume of the extraction solution in order to select the optimal volume for protein extraction, the homogenisation beads, the parts of the snail used and the snail species. All information about the protocols is summarized in Table 1. Finally, we applied the optimised protocol on the remaining snail samples in an attempt to validate the selected protocol.

**Ethanol-stored snail specimens.** A second protocol was developed for ethanol-stored samples specifically. Of the 84 specimens preserved in ethanol, 13 were used for protocol optimisation. Each dissected foot was rinsed twice in distilled water then dried overnight at 37°C, prior to homogenisation in order to remove any ethanol as previously described [40]. Three protocols, where we varied the amount of extraction solution or the grinding agent (glass beads ≤ 106μm or tungsten beads), were then compared for the preparation of snail samples. All the protocols are described in Table 1. Homogenisation was carried out in the same way as for the frozen specimens described above.

## Sample loading on the target plate and MALDI-TOF MS settings

All homogenates were centrifuged at 2.000g for 30 seconds and 1 μl of each supernatant was spotted onto a steel target plate (Bruker Daltonics) in quadruplicate. One microlitre of a CHCA matrix suspension composed of saturated alpha-cyano-4-hydroxycinnamic acid (Sigma, Lyon. France), 50% acetonitrile (v/v), 2.5% trifluoroacetic acid (v/v) (Aldrich, Dorset, UK) and HPLC grade water [38,40,41] was directly spotted onto each sample on the target plate to allow co-crystallisation, then was air-dried at room temperature before being inserted

into the MALDI-TOF MS instrument (Bruker Daltonics, Germany) for analysis. Mass spectrometry analysis was carried out using Microflex LT MALDI-TOF mass spectrometry (Bruker Daltonics machine) with Flex Control software (Bruker Daltonics). Measurements were performed in the linear positive-ion mode [26] within a mass range of 2–20 kDa. Each spectrum corresponded to ions obtained from 240 laser shots performed in six regions of the same spot. The spectrum profiles were visualized using Flex analysis, version 3.3, exported then to the Biotyper version 3.0 software and ClinProTools v.2.2 for analysis.

## Spectral analysis and creation of the reference database

The Composite Correlation Index (CCI) tool (MALDI-Biotyper v3.0. software, Bruker Daltonics) was used to assess spectra variations within each sample group according to the protocol tested, as previously described [42]. CCI was computed using the standard settings of the mass range 3000–12000 Da, resolution 4, 8 intervals and autocorrelation off. Higher correlation values (expressed by mean ± standard deviation [SD]) reflect higher reproducibility of MS spectra (A CCI match value of 1 represents complete correlation, whereas a CCI match value of 0 represents an absence of correlation) [42]. The MS peaks and the intensity calculation of both the head and the foot were analysed using ClinProTools v.2.2 software.

The spectrum profiles obtained from snail species were visualized with Flex analysis v.3.3 then exported to ClinProTools v.2.2 software packages (Bruker Daltonics, Germany) for data processing (smoothing, baseline subtraction) [37,43]. Intra-species reproducibility and inter-species specificity were assessed by comparing and analysing the spectral profiles obtained from the four spots of each individual snail specimen. Spectrum quality was validated by evaluating its intensity, the smoothness of the peaks, the flatness of the baseline, and its reproducibility compared to other MS spectra in the same category. Poor-quality MS spectra (Intensity<3.000 arbitrary units [a.u.], the presence of background noise) were excluded from the analysis. A dendrogram was performed using MALDI-Biotyper software v.3.0 to visualize the heterogeneity level of MS spectra from specimens of different species (hierarchical clustering of the mass spectra). To evaluate the impact of the storage method on the quality of the MS spectra, a comparison of MS spectra of 4 specimens from each species (*Bi. pfeifferi*, *Bu. truncatus*, *Bu. forskalii*) preserved either in ethanol or at -20˚C was performed using the MS dendrogram (MALDI-Biotyper v.3.0) and the principal component analysis (ClinProTools software). The mean of peaks intensity and the number of peaks were also calculated for these MS spectra. Good quality MS spectra (high peak intensity and reproducibility) from each species conserved either in ethanol or at -20˚C were selected and added to the database after being unambiguously molecularly identified. The design of the study is summarized in the flowchart represented in Fig 2.

## Blind test for study validation

The study was validated through a blind test (MALDI-Biotyper software v.3.0, Bruker Daltonics) using frozen and ethanol-stored snail specimens, with the exception of those used as MS reference spectra. The results are presented with Log-score values (LSVs) that correspond to the degree of homology between the query and the MS reference spectra. This value can range from 0 to 3. The spectrum with the highest LSV among the four spots was selected as a valid identification [40].

## DNA extraction and molecular studies of snails

Each remaining half-snail body was rinsed with distilled water and dried on sterile filter paper. The genomic DNA was extracted using EZ1 DNA Tissue kit (Qiagen) following the

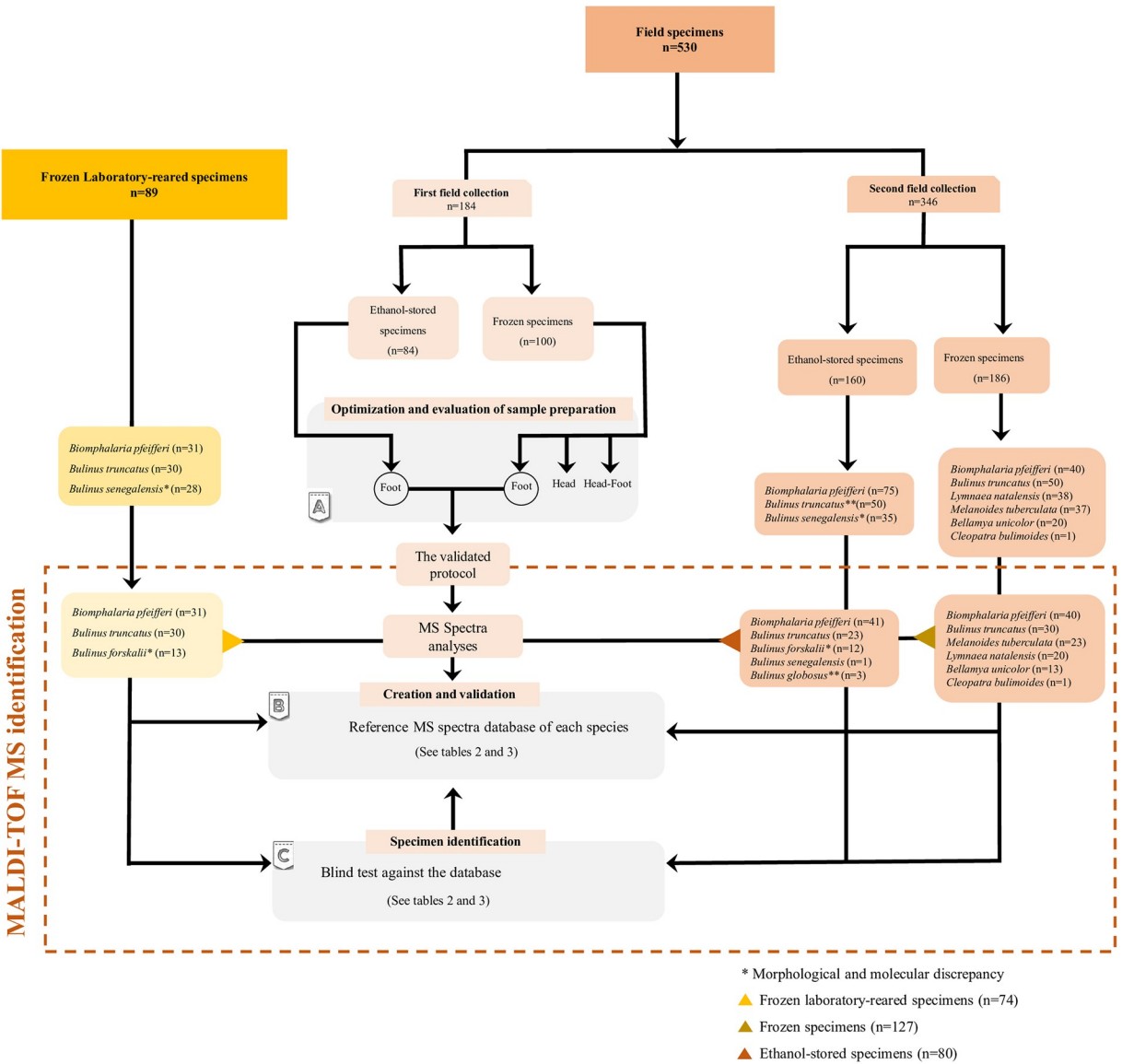

**Fig 2. Flowchart showing the study design and all samples collected from the first and second field collection [2018–2019] and those used for each step in the study.** The study design is divided into three sections: (A) protocol optimisation, (B) database creation and (C) the validation with a blind test. The number and the species of field and laboratory-reared freshwater snails are mentioned.

manufacturer's recommendations. Each snail sample was incubated at 56°C overnight in 180 µl of G2 lysis buffer (Qiagen Hilden, Germany) and 20µl proteinase K (Qiagen Hilden, Germany). The DNA extraction was then carried out using the EZ1 BioRobot extraction device (Qiagen Hilden, Germany). Genomic DNA from each sample was eluted with 100 µl of Tris-EDTA buffer (Qiagen) and was preserved at -20°C to be used for molecular analysis. For the molecular identification of snail species, DNA samples obtained from the specimens used as MS reference spectra were subjected to standard PCR in an automated DNA thermal cycler (Applied Biosystems, 2720, Foster City, USA). Two genes (*COI*, *16S*) were partially sequenced. A 710 base-pair region of the cytochrome c oxidase subunit I gene *(COI)* was amplified using Folmer's universal *COI* barcoding primers (LCO1490, HCO2198) [44] and for *16S*, the *16S*ar-L and *16S*br-H primers were used targeting 550 bp of the *16S* gene [45]. The amplified

products were visualised on 1.5% agarose gel stained with SYBR Safe then purified using a Macherey Nagel (NucleoFast 96 PCR, Düren, Germany) plate. Sequencing was performed using the BigDye Terminator v1.1, v3.1 5x Sequencing Buffer (Applied Biosystems, Warrington, United Kingdom) and run on an automated sequencer. Sequence chromatograms were assembled and edited using Chromas Pro1.77 (Technelyium Pty. Ltd, Tewantin, Australia). The sequences obtained were used to perform BLAST searches [46] via the National Center for Biotechnology Information (NCBI) GenBank database and were then aligned using (MEGA7) [47]. A phylogenetic tree was constructed and edited using the maximum likelihood method with model selection determined by MEGA7 and FigTree 1.4.2 respectively [48,49]. Statistical support for internal branches of the trees was evaluated by bootstrapping with 1000 iterations.

### Molecular screening for *Schistosoma* infection

Ethanol-preserved specimens were screened for the presence of parasites using primers (Smcyt748F-Smcyt847R) and a probe (Smcyt785T FAM) targeting a *COI* sequence of *S. mansoni* [50]. The repetitive *DRA1* sequence of *S. haematobium* (ShPCR) was amplified using primers (Sh-FW and Sh-RV) [51] and a probe from Cnops [52]. Quantitative PCRs were performed on the extracted DNA using the CFX96 Touch detection system (Bio-Rad, Marnes-la-Coquette, France) with the Light Cycler Probes Master (Indianapolis, IN USA). Our qPCR reaction mix included 5 µl of DNA, 10 µl Master Mix, 3.5 µl sterile distilled water and 0.5 µl of each of the primers and probe. For each qPCR plate, negative and positive controls were used. *Schistosoma haematobium* and *S. mansoni* previously obtained from snails infected with these species were used as positive controls while the negative control consisted of 20 µl of the qPCR reaction mix without any DNA.

## Results

### Morphological identification of snails

In Senegal, a total of 530 freshwater snail specimens (from 2018 to 2019) were collected and morphologically identified as *Bi. pfeifferi* (n = 239), *Bu. truncatus* (n = 127), *Bu. senegalensis* (n = 68), *L. natalensis* (n = 38), *M. tuberculata* (n = 37), *Be. unicolor* (n = 20) and *Cleopatra bulimoides* (n = 1). In addition, 89 laboratory-reared specimens including *Bi. pfeifferi* (n = 31), *Bu. truncatus* (n = 30), and *Bu. senegalensis* (n = 28) were also used for the study. Photographs of each species are presented in Fig 1.

### Standardisation of the MALDI-TOF MS protocol on frozen specimens

Snail MS profiles from three specimens of *Bi. pfeifferi* per protocol were compared using Flex Analysis based on spectral quality. By visual inspection, the MS spectra generated by protocol H2 (head, glass beads and 40µl Mix), protocol FH (foot-head, glass beads and 30µl Mix), and protocol F (foot, glass beads and 30µl Mix) were of higher quality than those of protocol H3 (head, glass beads and 30µl Mix) and protocol H1 (head, tungsten beads and 40µl Mix) (Fig 3A). The visual comparison of MS profiles using an unsupervised statistical test (PCA, ClinProtools software) revealed that the dots corresponding to MS spectra from H2, FH, F are separated from those of H1, H3 (Fig 3B). Using ClinProtools, the contributions of PC1, PC2, and PC3 to the generation of profile in a percentage the plot of the variance explained were approximately 38.18%, 16.81%, and 11.36%, respectively as presented in the following S1 Fig. To bring more robustness to the analyses, MS spectra reproducibility for each protocol was objectified using a CCI matrix. The highest CCI (Fig 3C) was obtained for protocol H2 (mean±SD:

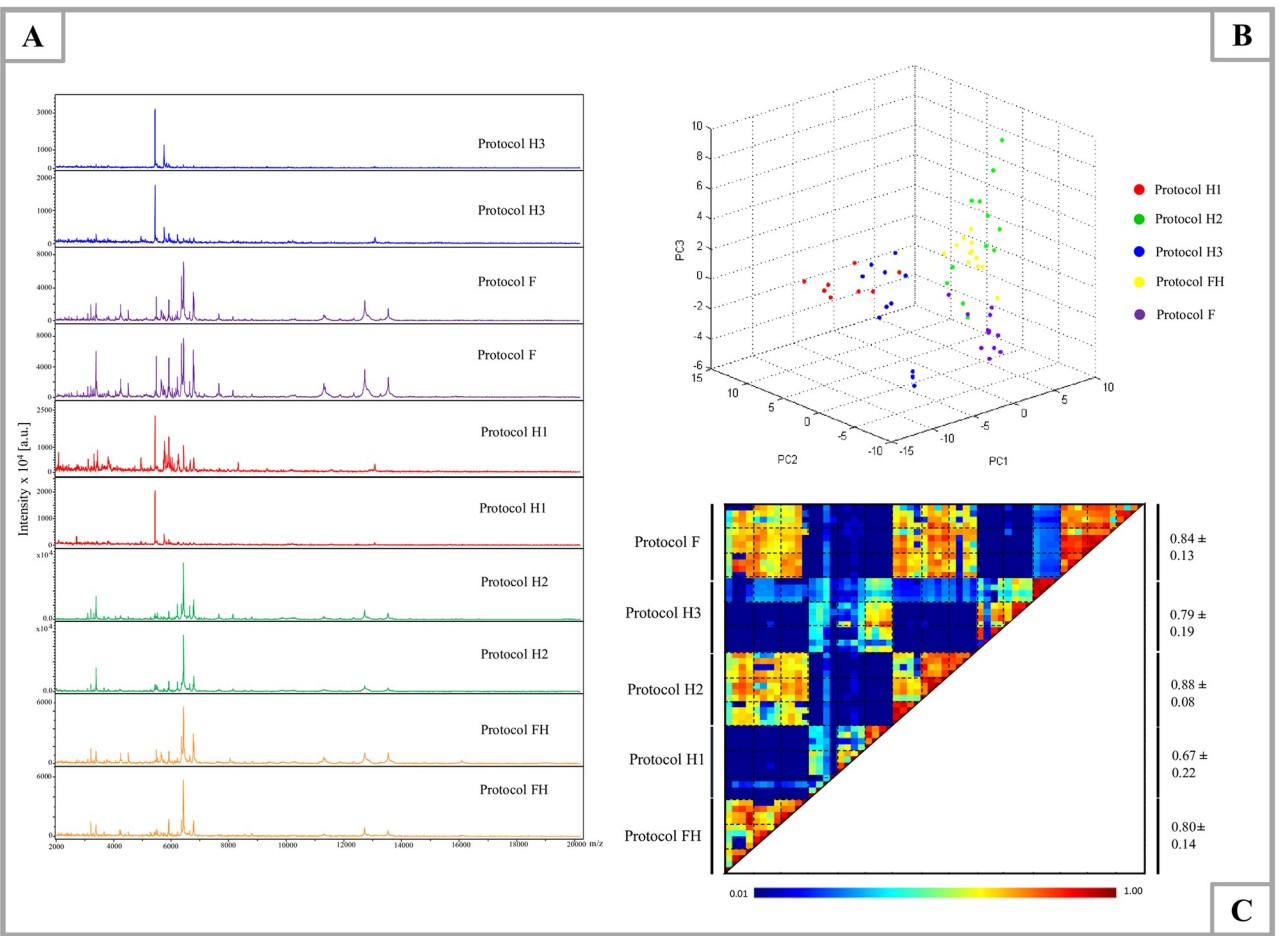

**Fig 3. Comparison of MALDI-TOF MS spectra of *Bi. pfeifferi* using different protocols (H: head, FH: foot-head, F: foot).** (A) Representative MS spectra from *Bi. pfeifferi* head (H1, H2, H3), foot-head (FH) and foot (F). (B) MALDI-TOF MS spectra distinction from *Bi. pfeifferi* between protocols compared by principal component analysis using ClinProTools v.2.2; protocol H1(40µl Mix+ tungsten beads), protocol H2 (40µl Mix+ glass beads), protocol H3 (30µl Mix+ glass beads), protocol FH (30µl Mix+ glass beads) and protocol F (30µl Mix+ glass beads). (C) Evaluation of MS spectra reproducibility generated by the five protocols using composite correlation index (CCI). The MS parameters: intensity ≥ 3.000 [a.u.], absence of background noise and the reproducibility of MS spectra. a.u.: arbitrary units; *m/z*: mass-to-charge ratio. The protocol F was selected to build the MS reference database and perform the analysis.

0.88±0.08) and protocol F (mean±SD: 0.84±0.13) than other protocols. The comparison of the number of MS peaks and intensity of specimens corresponding to the head (H2) and (F) showed that the mean intensity and number of MS peaks generated from *Bi. pfeifferi* head (mean intensity: 224.0342, number of MS peaks: 87) are higher than those generated from the foot (mean intensities: 173.0325, number of MS peaks: 74) (S1 Table). For further MALDI-TOF MS analysis, we selected protocol F because the foot generally harbours the parasites and could potentially be used for the detection of *Schistosoma*.

## MS identification of frozen snail species

In order to assess intra-species reproducibility and inter-species specificity, 127 of the 186 frozen snail specimens were randomly selected, including *Bi. pfeifferi* (n = 40), *Bu. truncatus* (n = 30), *M. tuberculata* (n = 23), *L. natalensis* (n = 20), *Be. unicolor* (n = 13), and *C. bulimoides* (n = 1) (Fig 2). 74/89 laboratory-reared specimens belonging to *Bi. pfeifferi* (n = 31), *Bu.*

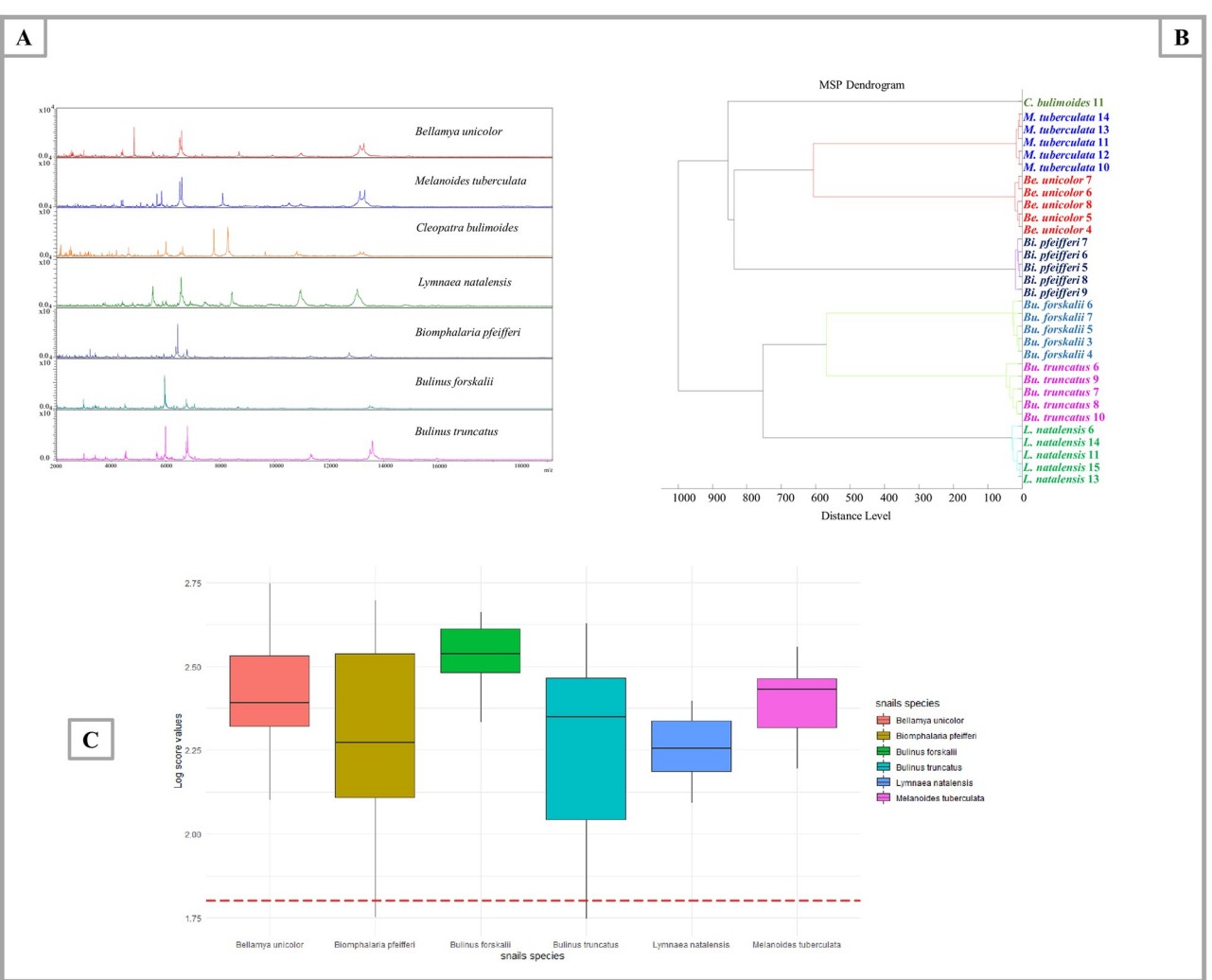

**Fig 4. MALDI-TOF MS spectra obtained from different snail species to create the database.** (A) Spectral alignment of seven snail species showing discriminative peaks using Flex Analysis software; smoothed spectra with baseline subtracted. (B) dendrogram of MALDI-TOF MS spectra of snail species collected in Senegal. Cluster analysis was performed using Biotyper software v.3.0. (C) Graphical representation showing the log-score values classification according to the following species: *Bu. forskalii*, *Bu. truncatus*, *M. tuberculata*, *Be. unicolor*, *L. natalensis* and *Bi. pfeifferi*.

*truncatus* (n = 30) and *Bu. senegalensis* (n = 13) were also used for MALDI-TOF MS analysis using the selected protocol F (Fig 2). Within the mass range of 2 to 20 kDa, the MS spectra for each species of snails were shown to be specific with strong signal intensities (Fig 4A). The MS protein profiles of one to five specimens of each of the snail species were randomly selected and used to create a dendrogram (Fig 4B). Cluster analysis reported that the MS profiles of each species (with confirmed DNA sequences) have a specific spectral fingerprint, as the specimens of the same species were clustered in the same branches. A database was then created using MALDI-Biotyper 3.0 and upgraded with MS spectra of high quality. To determine whether MALDI-TOF MS is an efficient way of discriminating between the MS profiles species, 166 specimens including laboratory-reared and field specimens were queried against the database, which includes seven molecularly identified species: *Bu. truncatus* (9), *Bi. pfeifferi* (6), *Bu. forskalii* (4), *Be. unicolor* (2), *L. natalensis* (3), *M. tuberculata* (2), and *C. bulimoides* (1). The blind test against the database showed a correct identification for all specimens

**Table 2. Laboratory-reared and field species selected to create a MALDI-TOF MS reference database, identified molecularly using *COI*.**

| Morphological ID | Collection site | N˚-specimens [tested] | Good quality spectra | Molecular identification (n = sequences) % of identity | Reference spectra database | Blind test | Snail species ID by MS | Score range |
|---|---|---|---|---|---|---|---|---|
| *Bi. pfeifferi* € & | Richard Toll | 71 | 66/71 | *Bi. pfeifferi* (6) 100% | 6 | 60/60 | *Bi. pfeifferi* | [1.751–2.696] |
| *Bu. truncatus* € & | | 60 | 59/60 | *Bu. truncatus* (9) 100% | 9 | 50/50 | *Bu. truncatus* | [1.746–2.628] |
| *L. natalensis* & | | 20 | 18/20 | *L. natalensis* (3) 99.32% | 3 | 15/15 | *L. natalensis* | [2.092–2.398] |
| *Be. unicolor** & | | 13 | 13/13 | *Be. capillata* (2) 97.04% | 2 | 11/11 | *Be. unicolor* | [2.101–2.748] |
| *C. bulimoides* & | | 1 | 1 | *C. bulimoides* (1) 89.63% | 1 | - | - | - |
| *M. tuberculata* & | | 23 | 23/23 | *M. tuberculata* (2) 96.13% | 2 | 21/21 | *M. tuberculata* | [2.194–2.558] |
| *Bu. senegalensis** € | Niakhar | 13 | 13/13 | *Bu. forskalii* (4) 99.09% | 4 | 9/9 | *Bu. forskalii* | [2.333–2.662] |
| Total | | 201 | 193 | 27 | 27 | 166 | | |

€ = Laboratory-reared snails;

* = Discrepancy between morphological and molecular identification;

& = Field specimens

(100%) at the species level with LSV scores between 1.746 and 2.748 which are presented in Fig 4C and Table 2. The median of log-score values is 2.362 and the mean of LSVs is 2.313 ±0.238. Snail MALDI-TOF MS database is publicly accessible and can be downloaded with the following DOI number: https://doi.org/10.35088/f605-3922.

## Standardisation of the MALDI-TOF MS protocol for ethanol-stored specimens

*Biomphalaria pfeifferi* feet were used for each protocol and submitted to MALDI-TOF MS analysis. Protocols F1 and F2 both yielded MS spectra of low quality (intensity and reproducibility) compared to protocol F3. The principal component analysis PCA (Fig 5A) revealed a clear distinction between protocol F3 and protocols F2 and F1; showing that the MS spectra of protocol F3 are visibly better (high peak intensities) than the MS spectra obtained by protocols F1 and F2. Using ClinProtools, the contributions of PC1, PC2, and PC3 to the generation of profile in a percentage plot of the variance explained were approximately 35.65%, 26%, and 17.39%, respectively as presented in the following S2 Fig.

Spectral profile analysis using Flex Analysis indicated that MS spectra from protocol F3 were of good quality, but some specimens showed slightly divergent profiles (Fig 5B). Eighty specimens out of 160 were used to validate protocol F3 (Fig 2). Of 80 MS spectra analysed by Flex analysis, 57 MS spectra were of good quality. The cluster analysis (Fig 5C) indicated that the randomly selected specimens from the species *Bi. pfeifferi* (n = 4), *Bu. truncatus* (n = 3), *Bu. globosus* (n = 3), *Bu. forskalii* (n = 3) and *Bu. senegalensis* (n = 1) were grouped together. The MS profiles of *Bi. pfeifferi*, *Bu. truncatus*, *Bu. globosus*, *Bu. forskalii* and *Bu. senegalensis* are presented in Fig 5D. Therefore, a specific database was created, comprising seventeen MS spectra: *Bi. pfeifferi* (n = 9), *Bu. truncatus* (n = 4), *Bu. forskalii* (n = 2), *Bu. globosus* (n = 1), and *Bu. senegalensis* (n = 1), and the remaining 40 MS spectra were used for the blind test (Table 3). Of the 40 specimens, 22/22 *Bi. pfeifferi*, 3/10 *Bu. truncatus*, 4/6 *Bu. forskalii* and 2/2 *Bu. globosus* were correctly identified at the species level (77.50%). Their LSVs ranged from 1.705 to 2.529 (Table 3). The principal component analysis (S3 Fig) unveiled a clear distinction

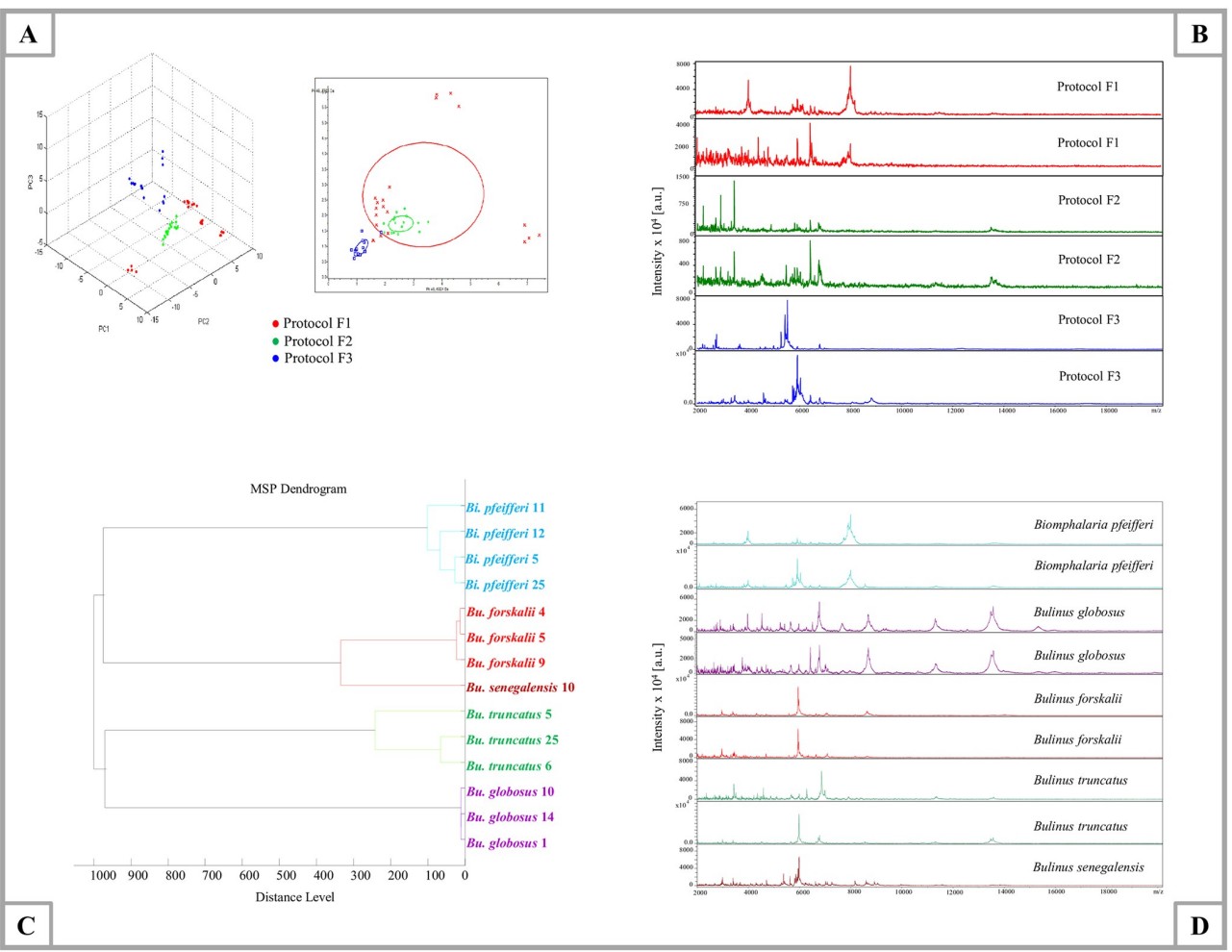

**Fig 5. Comparison of MALDI-TOF MS spectra from the foot of ethanol-stored specimens.** (A) Comparison of MALDI-TOF MS spectra of three different protocols on principal component analysis using ClinProTools v.2.2; protocol F1 (30μl Mix+ glass beads), protocol F2 (30μl Mix+ tungsten beads), and protocol F3 (15μl Mix+ glass beads). (B) Representative MS spectra of *Bi. pfeifferi* obtained for each protocol, analysed by Flex analysis. (C) Hierarchical clustering dendrogram of *Bi. pfeifferi*, *Bu. truncatus*, *Bu. globosus*, *Bu. forskalii*, and *Bu. senegalensis* performed by Biotyper software v.3.0. (D) Representative MALDI-TOF MS profiles of *Bi. pfeifferi*, *Bu. truncatus*, *Bu. globosus*, *Bu. forskalii* and *Bu. senegalensis* using protocol F3, performed using Flex Analysis Software. a.u.: arbitrary units; *m/z*: mass-to-charge ratio. The x-axis represents PC1 and the y-axis represents PC2.

between the MS spectra of specimens of the same species preserved either at -20˚C or in ethanol. The MS spectra comparison of both storage conditions using a dendrogram also showed that the MS spectra of frozen specimens of each species (*Bi. pfeifferi*, *Bu. truncatus* and *Bu. forskalii*) clustered separately from ethanol-stored MS spectra of the same species (S4 Fig). Accordingly, a specific database is necessary for each storage condition in order to get a correct identification for frozen and ethanol-stored specimens of each species. The mean spectra intensities and the number of MS peaks of *Bi. pfeifferi*, *Bu. truncatus* and *Bu. forskalii* were higher for frozen specimens than ethanol-stored specimens of the same species. All the obtained values of MS spectra intensities and the number of MS peaks are presented in S3 Table. In the obtained dendrogram (S4 Fig), we noticed that the MS spectra of frozen specimens were so different from the ones in alcohol of the same species that they were positioned on another branch completely different from the frozen ones and in some cases, they were so much different that they were closer to another species. Particularly, in this case, frozen

**Table 3. Identification results of ethanol-stored snail specimens including schistosomes detection results.**

| Morphological ID | Specimen collection | Tested samples | Good quality spectra | Molecular identification (n = sequences) % of identity | Reference spectra | Blind test | Species ID by MS | Score range | Score average | *S. haematobium* | *S. mansoni* |
|---|---|---|---|---|---|---|---|---|---|---|---|
| *Bi. pfeifferi* | 75 | 41 | 31/41 | *Bi. pfeifferi* (9) 100% | 9 | 22/22 | *Bi. pfeifferi* | [1.705–2.371] | 1.974 | - | 1/41 (2.43%) |
| *Bu. truncatus*\* | 50 | 26 | 17/26 | *Bu. truncatus* (8) 100% | 4 | 3/10 | *Bu. truncatus* | [1.711–1.734] | 1.719 | 3/39 (7.69%)- | - |
| | | | | *Bu. globosus* (1) 100% | 1 | 2/2 | *Bu. globosus* | [1.878–1.886] | 1.882 | - | - |
| *Bu. senegalensis*\* | 35 | 13 | 9/13 | *Bu. senegalensis* (1) 100% | 1 | 4/6 | - | - | - | 1/39 (2.56%)- | - |
| | | | | *Bu. forskalii* (5) 99.09% | 2 | | *Bu. forskalii* | [1.850–2.529] | 2.071 | - | - |
| Total | 160 | 80 | 57/80 | 24 | 17 | 31/40 | - | - | - | 4/39 (10.25%) | 1/41 (2.43%) |

*Discrepancy between morphological and molecular identification

*Bu. truncatus* MS spectra were closer to the MS spectra of *Bu. forskalii*. Consequently, this reflects a deep difference between these two storage conditions.

## Molecular identification of snails

Twenty-seven specimens were selected for database creation. We successfully obtained 27 sequences of high quality of different species using *COI* gene: 9 *Bu. truncatus*, 6 *Bi. pfeifferi*, 4 *Bu. senegalensis*, 3 *L. natalensis*, 2 *Be. unicolor*, 2 *M. tuberculata* and 1 *C. bulimoides*. NCBI BLAST analysis indicated that snails morphologically identified as *Bi. pfeifferi* (ID score 100%: AF199099.1), *Bu. truncatus* (ID score 100%: MT272328.1), *L. natalensis* (ID score 99.32%: HG977206.1), *M. tuberculata* (ID score 96.13%: MK879274.1) matched with their respective homologous sequences available in GenBank. However, the sequences obtained from snails that had been morphologically identified as *Bu. senegalensis* were more closely related to the sequences deposited in GenBank as *Bu. forskalii* sequences (ID score 99.09%: AM286310.4). It should also be noted that the sequences obtained from snails that had been morphologically identified *Be. unicolor* matched with sequences available in GenBank as *Bellamya capillata* (ID score 97.04%: JX489247.1). The obtained sequence of *C. bulimoides* (Morelet, 1860) matched with an identity of 89.63% with the GenBank reference sequence for *C. bulimoides* originated from Egypt (KF412769.1) (Table 2).

For ethanol-stored specimens, we obtained *COI* sequences for 24 specimens belonging to species of the genus *Biomphalaria* and *Bulinus*: 17 were used for database creation and the remaining specimens (n = 7) were used for the validation of the morphological identification. All nine *Bi. pfeifferi* were correctly identified (ID score 100%: AF199099.1). Of the nine specimens morphologically identified as *Bu. truncatus*; eight sequences matched with *Bu. truncatus* sequences deposited in GenBank (ID score 100%: MT272328.1) and one sequence was identified as *Bu. globosus* (ID score 100%: AM921808.1). As for specimens identified morphologically as *Bu. senegalensis*, six sequences were obtained: one sequence matched with *Bu. senegalensis* (ID score 100%: KJ157485.1) and the five remaining sequences were closely related to *Bu. forskalii* (ID score 99.09%: AM286310.4) (Table 3).

The phylogenetic tree constructed on the basis of the *COI* fragment sequences (Fig 6) highlighted that *COI* sequences of seven species (*Bi. pfeifferi*, *Bu. truncatus*, *Bu. globosus*, *Bu. forskalii*, *Bu. senegalensis*, *L. natalensis* and *M. tuberculata*) clustered with their homologous

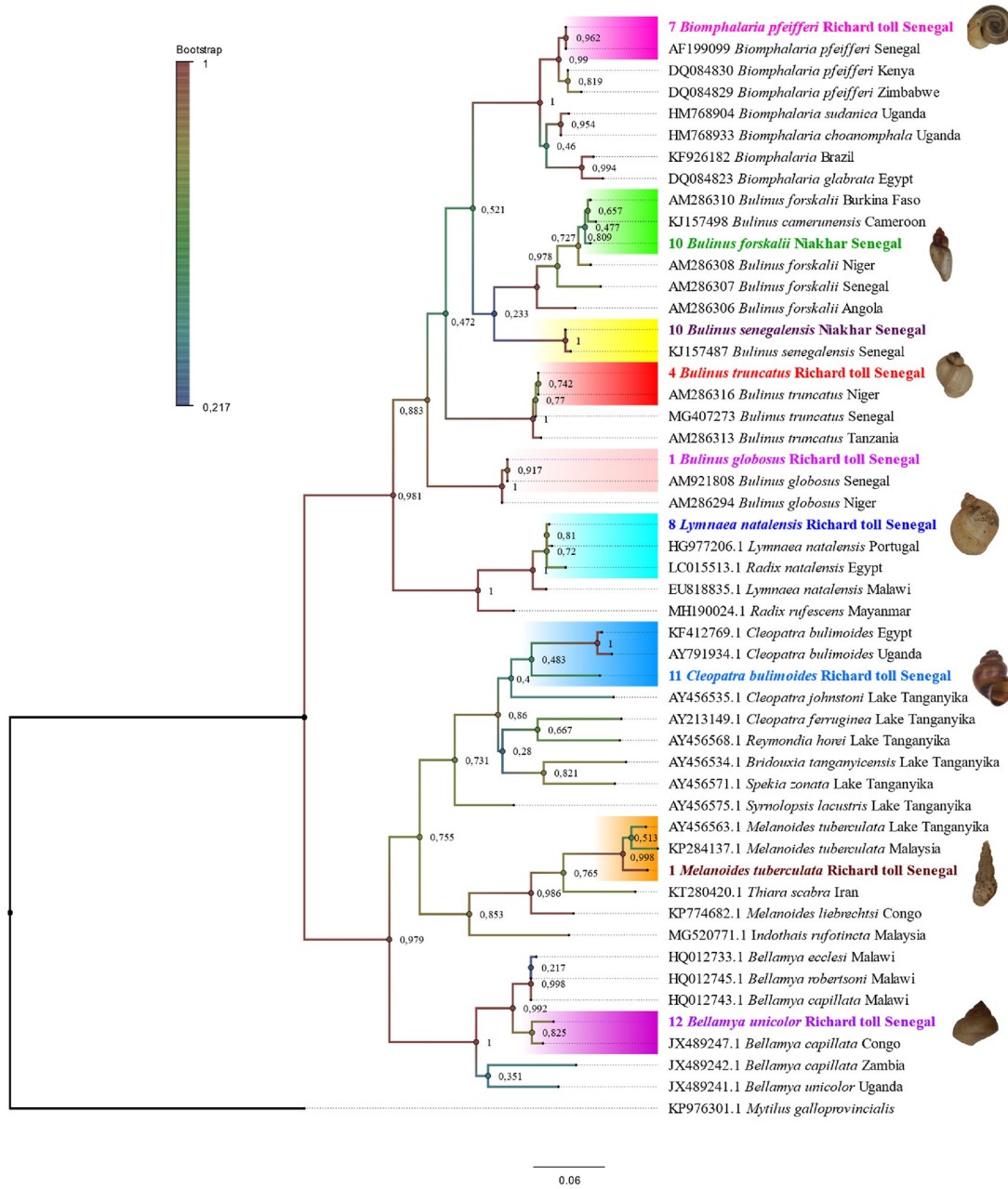

**Fig 6. Phylogenetic tree showing the position of nine snail species (*Bi. pfeifferi*, *Bu. truncatus*, *Bu. forskalii*, *Bu. senegalensis*, *Bu. globosus*, *Be. unicolor*, *M. tuberculata*, *L. natalensis*, and *C. bulimoides*) used in this study.** The tree was constructed using the maximum likelihood method based on the Kimura 2-parameter distance (MEGA7). The values on the branches are bootstrap support values based on 1000 replications. Branches are colour-coded to the bootstrap values. The identity of each taxon is colour-coded conforming to the species.

sequences available in GenBank, with high bootstrap values. *Bellamya unicolor* sequences clustered with *Be. capillata* sequences while *C. bulimoides* clustered separately and formed a distinct branch. In order to confirm the molecular identification of both specimens that had been morphologically identified as *Be. unicolor* and *C. bulimoides*, a fragment of the *16S* gene was sequenced. The BLAST results showed that sequences of *Be. unicolor* (n = 2) matched with

sequences of *Bellamya jeffreysi* (ID range score 96.85–97.03%: FJ405702.1) and the sequence of the *C. bulimoides* (n = 1) was designated as *Cleopatra johnstoni* (ID score 95.71%: KF412769). These results (S2 Table) reveal that identification at the species level is still not clear with the second gene (*16rRNA*). All sequences (*COI* and *16rRNA*) were deposited in GenBank and all FASTA sequences are available in S1 Data.

### Detection of schistosomes in ethanol-stored snails

We performed a qPCR for the detection of *S. mansoni* and *S. haematobium* species complex. Eighty snail specimens belonging to *Bi. pfeifferi* (n = 41), *Bu. truncatus* (n = 26), and *Bu. senegalensis* (n = 13) were screened. Of the ethanol-stored snails tested, 4/39 (10.25%) *Bulinus* (1 *Bu. senegalensis* and 3 *Bu. truncatus*) were positive by qPCR for *S. haematobium* and 1/41 (2.43%) *Bi. pfeifferi* were positive for *S. mansoni* (Table 3). The determination of the infectious status based on the comparison of MS spectra of both infected and uninfected specimens belonging to *Bi. pfeifferi*, *Bu. truncatus* and *Bu. senegalensis* was not performed because of the low number of infected specimens.

## Discussion

This study presents a rapid, low-cost, and reliable approach for the routine identification of freshwater snail specimens including intermediate hosts of schistosomes by optimising the MALDI-TOF MS sample preparation protocol for both frozen and ethanol-preserved specimens. Recently, the MALDI-TOF MS technique has emerged in medical entomology and has proven its effectiveness in the context of identifying various species of arthropods such as fleas [29], ticks [27,53–56], sand flies [57,58], mosquitoes [28,59–63], culicoides [43], tsetse flies [64], triatomines [65] and bed bugs [66] using proteins extracted from different body parts that generate MS spectra of high quality. Furthermore, this proteomic approach has also been applied to malacology as Stephan *et al.* were the first to report the MALDI-TOF MS use for identification of marine bivalve molluscs specifically the scallops [30].

The quality of MS spectra is impacted by several parameters such as the storage conditions, the adjustment of the extraction solution volume used for protein extraction, the grinding method, and the selected body part [38]. In the first step of our study, we tested two kinds of disrupters: both tungsten beads and glass beads have proven to be useful and effective grinding methods in previous MALDI-TOF MS studies [38,40,66,67]. However, glass beads were reportedly the most commonly used disrupter, as they have been used for various arthropods identification [39,40,66], for mycobacteria identification, to disrupt the mycobacterial cell wall [68,69], and for fungal identification, as they were most efficient than steel beads for fungal proteins extraction [70]. The glass beads also appeared to be the most suitable for snail sample homogenisation, as they provided reproducible and intense MS spectra for snail species. As for the matrix preparation, we used alpha-cyano-4-hydroxycinnamic acid which has been successfully used for several arthropods identification [27,40,42,66]. As well, it has shown its performance in clinical microbiology. In particular, the identification of many microorganisms such as mycobacteria, bacteria, yeasts, and fungi [71–74]. Besides, the use of this matrix was sufficient to yield MS spectra of high quality of different snail species. Therefore, we validated the use of alpha-cyano-4-hydroxycinnamic acid for matrix preparation. Otherwise, we would have tested different acids as sinapinic acid in case we have failed to get good signals with the best resolution. The use of such matrix preparation and glass beads have previously shown their performance and robustness in different conditions [39,40,68–71]. Consequently, in the present work, the combination of glass beads and matrix based on alpha-cyano-4-hydroxycinnamic acid revealed promising results for freshwater snail identification. In addition, we

standardised a protocol for frozen specimens. Two body parts, the head (H) and the foot (F), and a combination of foot and head (FH) were submitted for MALDI-TOF MS analysis in order to get a specific spectral signature for each species. The findings showed that MALDI-TOF MS spectra obtained from the snail foot (F), head (H) and foot-head (FH) protein extracts were adequate to accurately identify freshwater snails involved in schistosomiasis transmission. The protocol validation was confirmed using only the foot. Precisely, we selected the foot for sample preparation for two reasons, firstly, the primary sporocysts of *Schistosoma* generally occur in the foot and secondly, it was reported that this snail's body part was a good protein source compared to other snail tissues [36,75]. The intensity and number of peaks generated from the *Bi. pfeifferi* foot (S1 Table) used for protocol optimisation were slightly lower than those utilized for protocol validation (S3 Table). This can be explained by the fact that the samples manipulated for the first time had to be frozen and thawed many times in order to validate the morphological identification and MALDI-TOF MS essays. Frequent thawing could be responsible for protein degradation. The same observations were stated by Ouarti *et al.* (2020) [76]. The volume of the extraction solution is an important parameter to have MS spectra of good quality and as previously reported on arthropods [39,77] that the extraction solution volume differs from one arthropod to another and from one part of the same arthropod to another. For this reason, we have varied the volume of the extraction solution in order to select the volume that allows getting good quality and reproducible MS spectra. In our work, the optimisation of the extraction solution volume was not based on the weight of the snail's foot and head tissue. However, for more precision, in future projects on the detection of schistosomes within the intermediate hosts by MALDI-TOF MS, it will be crucial to refine the protocol by taking into consideration the weight of the snail's foot tissue. Overall, in our study, we showed that the MALDI-TOF MS analysis of snail's foot tissue allows a correct identification to the species level, without requiring any malacological skills.

Cluster analysis (Fig 4B) highlighted the distinction between snail species and the visual comparison of MS profiles (Fig 4A) revealed discriminative peaks between species. To create the database, we used first-generation laboratory-bred specimens to ensure that the MS spectra included in our database belong to non-infected snail specimens [78,79]. In addition, the validity of the database was assessed using a blind test, in which 100% of specimens from six species were correctly identified (Table 2). However, there were not enough *C. bulimoides* specimens available for a blind test. The identification of this species was only confirmed morphologically as *C. bulimoides* since no reference sequence existed in GenBank for *C. bulimoides* (Morelet, 1860) originated from Senegal. Based on the shell morphology, *C. bulimoides* was reported to be polytypic and highly varied species including several synonymies [33].

In our study, the LSV cut-off of 1.7 was proposed for accurate species-level identification of several freshwater snails. A comparable cut-off value was used in a study on the identification of cercariae using MALDI-TOF MS [25]. Many other entomological studies have used a closely comparable cut-off (1.8), which was shown to be linked with the correct identification of mosquitoes, fleas and ticks [28,29,41].

We obtained some discrepancies between molecular identification and the morphological identification carried out in Senegal. In fact, four sequences of specimens morphologically identified as *Bu. senegalensis* were more closely related to sequences deposited in GenBank as *Bu. forskalii* sequences (ID score 99.09%). This can be explained by the limited morphological distinctiveness within the *forskalii* group (*Bu. senegalensis* and *Bu. forskalii*) [80]. Based on conchological analyses, it has been reported that it might not be easy to separate the species (*Bu. senegalensis* and *Bu. forskalii*) on shell characteristics alone, despite their close resemblance [80]. Indeed, *Bu. senegalensis* can be distinguished from *Bu. forskalii* by the lack of any carination forming a shoulder on the upper whorls [32] whereas Mandahl-Barth pointed out

that this distinctive shoulder angle was not always found in *Bu forskalii*; it was either absent or faint which made it not easy to be observed [81].

In order to upgrade the database with MS spectra of unambiguously identified specimens, we used the mitochondrial cytochrome oxidase subunit I gene, based on previous molecular studies reporting that the *COI* gene presents considerable divergent sequences making it reliable for discriminating the *Bulinus* species [82,83]. A previous study demonstrated that these morphologically similar species are differentiated by molecular methods, indicating the advantage of targeting the partial *COI* gene to accurately identify species within the same group [18]. *COI* sequences are also available for both species (*Bu. forskalii* and *Bu. senegalensis*) in the Gen-Bank database. Based on a fragment of the *COI* gene, we confirmed that the species identified morphologically as *Bu. senegalensis* were in fact *Bu. forskalii*.

Additionally, for morphologically identified samples of *Be. unicolor*, we found a discrepancy with the molecular identification. Two *Be. unicolor* sequences matched with a *Be. capillata* sequence deposited in GenBank (ID score 97.04%). It has been previously reported that it is challenging to separate *Be. capillata* from *Be. unicolor* morphologically [33]. The misidentification could be due to overlapping characteristics of the shell, which can be fairly variable in *Bellamya* species [84]. Based on morphology, we confirmed that our *Bellamya* specimens are *Be. unicolor* based on formally identified shell characteristics. To the best of our knowledge, *Be. capillata* has never been found in Senegal so far but it is present in Eastern (Tanzania) and Central African countries [85, 86]. In order to resolve the problem of discordance between morphological and molecular identification, we used the *16S* rRNA gene as a second reference for species validation. The BLAST results showed that the sequence of the specimen morphologically designated as *C. bulimoides* matched with C. johnstoni, with an identity of 95.71%. This could be explained by the fact that *C. bulimoides* is a highly varied species [33] and the *C. bulimoides* (originated from Senegal) used in our study was not available in the GenBank. The conspecifity of this highly varied species needs to be parsed out genetically in further studies [33]. In contrast, sequences of the specimens morphologically identified as *Be. unicolor* matched with *Be. jeffreysi* (ID range score 96.85–97.03%: FJ405702.1). As previously reported, the taxonomy of the viviparid species has been impaired by a lack of data concerning their evolutionary taxonomy, which leads to gaps in identification at the species level [86,87]. Moreover, a prior study conducted on genetic diversity and the phylogeny of the *Bellamya* species collected from different lakes in Africa revealed that the genetic diversity of both *16S* and *COI* was very low within clade variation. This study also reported that the characterisation of the *Bellamya* species located in African lakes using DNA barcoding could not be sufficient because there is less molecular divergence than morphological divergence [86]. Consequently, we support the conclusion made by Sengupta *et al*. and Mandahl-Barth [32] based on morphology, and confirmed the identification of the species of *Bellamya unicolor* using conchological criteria. Here, we also raised the question about the reliability of molecular tools for the identification of the *Bellamya* species. For this first section of the study, we showed that MALDI-TOF MS is a reliable tool for the identification of different frozen snail species. It could be used to circumvent molecular limitations such as the long time required for sample preparation, high-cost reagents, and even the lack of reference data.

In the second step of the study, we assessed the efficiency of MALDI-TOF MS in identifying three snail species: *Bi. pfeifferi*, *Bu. senegalensis* and *Bu. truncatus* collected in the field and stored in ethanol. The use of ethanol is crucial since the majority of analytical laboratories are far from collection sites and it is known to be the ideal storage method, particularly in low or middle-income countries. Additionally, it is cheaper than frozen methods and preserves specimens flexible, allowing for later morphological study. However, previous studies revealed that the use of frozen and fresh specimens arthropods provide reproducible and better MS spectra,

compared to those stored in alcohol [41,88], probably due to ethanol causing alterations to the protein structure [89]. Nevertheless, it was recently shown that these limitations could be circumvented in ticks by applying a dealcoholisation protocol, which is now our laboratory reference protocol [40]. In this study, we used the same body part (the foot) used to identify frozen specimens and followed the dealcoholisation steps proposed by Diarra *et al.* [40] as a means of perfusing out any residual alcohol from the foot tissue that could interfere with subsequent MALDI-TOF MS sample preparation. It is important to note that the MS profiles were of better quality when we used glass beads as an agent disrupter, while the tungsten beads resulted in MS spectra of low quality. For this reason, we validated glass beads as the most suitable means for sample disruption for ethanol-stored specimens (Fig 5A). The results of the blind tests of the alcohol-preserved specimens (Table 3) revealed that 22/22 of *Bi. pfeifferi*, 4/6 of *Bu. forskalii*, 2/2 of *Bu. globosus* and only 3/10 of *Bu. truncatus* were correctly identified with reliable LSVs (LSV $\geq$ 1.7). This might be attributed to the limited number of specimens tested in the blind test and used for the creation of the database. We agree with previous studies reporting the effect of ethanol 70% on the reproducibility of MALDI-TOF MS spectra [90, 91] and the difference that exists between MS spectra of frozen and ethanol-stored specimens for each species [88]. In this part of the study, we concluded that MALDI-TOF MS can be used for the identification of snails stored in alcohol, but that requires the creation of a specific database with a large number of specimens preserved in ethanol to circumvent the bias of heterogeneity. On the other hand, we noticed differences at the MS profiles level of both frozen and ethanol-stored specimens of the same species. Specifically, the mean spectra intensities and the number of MS peaks of frozen specimens of *Bi. pfeifferi*, *Bu. truncatus* and *Bu. forskalii* were higher than those of specimens of the same species stored in ethanol (S3 Table). This difference may be due to the effect of ethanol on the quality of the MS spectra as previously reported in ethanol-stored lice, ticks, fleas and cercariae [25,40,53,67,77,88].

To assess whether the heterogeneity of the MS profiles of each species was due to schistosomes infection, the pathogen screening was performed. Unfortunately, we could not compare MALDI-TOF MS spectra from *Schistosoma*-infected snails to the uninfected ones because the number of infected specimens was not enough to create a specific database for *Schistosoma*-infected snails and also perform a blind test in order to validate the results. Moreover, we could not perform a robust analysis using the principal component analysis (PCA) to distinguish infected and non-infected specimens because of the low number of MS good quality spectra from infected specimens; *Bi. pfeifferi* (n = 1), *Bu. truncatus* (n = 2), and *Bu. senegalensis* (n = 1). In view of the fact that so few samples were infected, it is unlikely that the infection was the source of the general heterogeneity. Therefore, we suppose that the organic preservation solution (ethanol) may be responsible for the MS profile modification of specimens as previously discussed in other studies [67,88]. Echinostomes and schistosomes are two genera of trematodes that utilize freshwater snails [92] as intermediate hosts. As stated by a recent study, echinostomes can infect some intermediate hosts of schistosomes, namely *Bulinus* and *Biomphalaria*, as their first intermediate host [93]. However, echinostomes were shown to be active competitors of schistosomes, limiting the ability of the latter to settle in the snail [94,95]. It was specifically reported that *E. caproni* limits the infection of *Bi. glabrata* by *S. mansoni* [96]. In our study, snail specimens belonging to *Biomphalaria* and *Bulinus* were screened for the presence of *Schistosoma* infection but these other parasites such as *Echinostoma* or Paramphistomidae [93,97] could infect the same species and may potentially impact the MS spectra. In future studies aimed at developing MALDI-TOF MS for the detection of schistosomes in snails, it would be interesting to better characterise the specimens and ensure the thorough investigation of parasites commonly associated with the tested snails. This current study is however focused on the application of MALDI-TOF MS technique in order to identify snail

populations including the species that play an indispensable role in the schistosomiasis transmission. Provided that the MALDI-TOF MS protocol has been standardized, it is necessary to assess the ability of MALDI-TOF MS to detect the infection within the intermediate hosts either by artificially infecting the freshwater snails (*Biomphalaria* or *Bulinus*) or collecting snails in *Schistosoma*-endemic areas in an attempt to help mapping the risk of disease transmission or to set-up long-term control strategies.

Overall, this high-throughput technique made it possible to identify both frozen and ethanol-preserved snail specimens, with minimal sample requirements and rapid availability of results compared to the current DNA based-identification.

## Conclusion

Our study shows, for the first time, that it is possible to use foot protein extracts for snail species identification as well as closely related species using MALDI-TOF MS. Our data indicate that MALDI-TOF MS is a robust and suitable alternative tool for the identification of several snail species preserved at -20°C or stored in ethanol, although the latter method does have some limitations. This high-throughput approach is rapid, low-cost, and could be used for closely related species belonging to the genus *Bulinus*, such as *Bu. senegalensis* and *Bu. forskalii* which are hardly differentiated morphologically. Its advantages make it a convenient approach for epidemiological studies in *Schistosoma*-endemic areas. In addition, the rapid identification of snail populations can provide a valuable advantage for the control of schistosomiasis. Since a MALDI-TOF MS machine is available in our research laboratory for microbiological and entomological purposes, it will also be used for future malacological surveys. Consequently, we will continue to update our MS database in collaboration with our research team and the parasitology laboratory in Dakar, Senegal. A recent study highlighted the potential of MALDI-TOF MS for the rapid identification of cercaria [25], so in the near future, it would be interesting to assess whether this innovative tool could successfully differentiate *Schistosoma*-infected specimens from non-infected specimens in order to use this approach for endemicity surveillance.

## Abbreviations

The accepted abbreviation for *Bellamya*, *Bulinus* and *Biomphalaria* is "*B.*" For clarity purposes, we chose to exceptionally abbreviate *Bulinus*, *Bellamya* and *Biomphalaria*: *Bu.*, *Be.*, *Bi.* respectively to distinguish them more easily in this manuscript.

## Supporting information

**S1 Fig. The dimensional image from PCA indicating the comparison of MALDI-TOF MS spectra from five different protocols (H1, H2, H3, FH, F) using frozen *Bi. pfeifferi*.** The figure shows the contribution of ten principal components to the profiling classification in plot of percentage explained variance of PC. Red dots: protocol H1, green dots: protocol H2, blue dots: protocol H3, yellow dots: FH and purple dots: F.
(TIF)

**S2 Fig. The dimensional image from PCA indicating the comparison of MALDI-TOF MS spectra from the foot of ethanol-stored *Bi. pfeifferi* specimens belonging to three different protocols.** The figure shows the contribution of eight principal components to the profiling classification in plot of percentage explained variance of PC. Red dots: protocol F1, green dots: protocol F2, blue dots: protocol F3.
(TIF)

**S3 Fig. MALDI-TOF MS spectra distinction from frozen and ethanol- stored *Bi. pfeifferi*, *Bu. forskalii*, *Bu. truncatus* by principal component analysis using ClinProTools 2.2.** (A) *Bi. pfeifferi*, (B) *Bu. forskalii*, (C) *Bu. truncatus.*
(TIF)

**S4 Fig. Comparison of MALDI-TOF MS spectra of frozen and ethanol-stored from four specimens of three species: *Bi. pfeifferi*, *Bu. truncatus* and *Bu. forskalii* using a dendrogram (Biotyper software v.3.0).** Frozen specimens are indicated in Bold and ethanol-stored specimens are not in Bold.
(TIF)

**S1 Data. *16S* sequences of *Be. unicolor* and *C. bulimoides*.** *COI* sequences of nine species including *Bi. pfeifferi*, *Bu. truncatus*, *Bu. forskalii*, *Bu. senegalensis*, *Bu. globosus*, *Be. unicolor*, *M. tuberculata*, *L. natalensis*, and *C. bulimoides.*
(TXT)

**S2 Data. CVs of the snail's foot and the head.**
(XLSX)

**S1 Table. The comparison of the mean intensity, number of MS peaks and the signal to noise from *Bi. pfeifferi* head (H2) and foot (F2).**
(DOCX)

**S2 Table. *16S* partial sequences obtained from species that were morphologically identified as *Be. unicolor* and *C. bulimoides*.**
(DOCX)

**S3 Table. Comparison of the mean intensity and the number of MS peaks of four specimens per species (*Bi. pfeifferi*, *Bu. truncatus* and *Bu. forskalii*) stored at -20˚C and in ethanol.**
(DOCX)

## Acknowledgments

Our thanks go to Lionel Almeras, Jean-Michel Bérenger, Younes Laidoudi and Hacene Medkour for their advice.

## Author Contributions

**Conceptualization:** Fatima Zohra Hamlili, Souleymane Doucouré, Babacar Faye, Cheikh Sokhna, Doudou Sow, Philippe Parola.

**Data curation:** Fatima Zohra Hamlili, Maureen Laroche, Adama Zan Diarra.

**Formal analysis:** Fatima Zohra Hamlili, Maureen Laroche, Adama Zan Diarra.

**Funding acquisition:** Doudou Sow, Philippe Parola.

**Investigation:** Fatima Zohra Hamlili, Fatou Thiam, Maureen Laroche, Adama Zan Diarra, Papa Mouhamadou Gaye, Cheikh Binetou Fall, Doudou Sow.

**Methodology:** Fatima Zohra Hamlili, Maureen Laroche, Adama Zan Diarra, Doudou Sow, Philippe Parola.

**Project administration:** Doudou Sow, Philippe Parola.

**Resources:** Fatou Thiam, Maureen Laroche, Doudou Sow, Philippe Parola.

**Software:** Fatima Zohra Hamlili, Maureen Laroche, Adama Zan Diarra.

**Supervision:** Philippe Parola.

**Validation:** Fatima Zohra Hamlili, Maureen Laroche, Adama Zan Diarra, Souleymane Doucouré, Doudou Sow, Philippe Parola.

**Visualization:** Fatima Zohra Hamlili, Philippe Parola.

**Writing – original draft:** Fatima Zohra Hamlili, Philippe Parola.

**Writing – review & editing:** Fatima Zohra Hamlili, Fatou Thiam, Maureen Laroche, Adama Zan Diarra, Souleymane Doucouré, Papa Mouhamadou Gaye, Cheikh Binetou Fall, Babacar Faye, Cheikh Sokhna, Doudou Sow, Philippe Parola.

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
