## [Decision Letter · Decision Letter 0]

15 Feb 2021

Dear Pr. Parola,

Thank you very much for submitting your manuscript "MALDI-TOF mass spectrometry for the identification of freshwater snails, including intermediate hosts of schistosomes in Senegal" for consideration at PLOS Neglected Tropical Diseases. As with all papers reviewed by the journal, your manuscript was reviewed by members of the editorial board and by several independent reviewers. In light of the reviews (below this email), we would like to invite the resubmission of a significantly-revised version that takes into account the reviewers' comments. In particular, Reviewer #1 raised major concerns about the MALDI-TOF-MS methodology and results. Reviewer #2 raised the important point of having difficulty in understanding the study design due to the large number of samples and different species/groups analyzed, and experiments performed, and suggested the addition of a flow chart to make the study design more clear to the readers.

We cannot make any decision about publication until we have seen the revised manuscript and your response to the reviewers' comments. Your revised manuscript is also likely to be sent to reviewers for further evaluation.

Sincerely,

Igor C. Almeida

Associate Editor

Sergio Costa Oliveira

Deputy Editor

Reviewer's Responses to Questions

**Key Review Criteria Required for Acceptance?**

**Methods**

-Are the objectives of the study clearly articulated with a clear testable hypothesis stated?

-Is the study design appropriate to address the stated objectives?

-Is the population clearly described and appropriate for the hypothesis being tested?

-Is the sample size sufficient to ensure adequate power to address the hypothesis being tested?

-Were correct statistical analysis used to support conclusions?

-Are there concerns about ethical or regulatory requirements being met?

Reviewer #1: The manuscript entitled: “MALDI-TOF mass spectrometry for the identification of freshwater snails, including intermediate hosts of schistosomes in Senegal” describes the optimization and application of MALDI-TOF to identify freshwater snails. Utilizing fresh frozen and ethanol-stored snail species and dissecting different parts, the authors have optimized a MALDI-TOF method for accurate identification of snail species.

Methods:

The objectives of the study are clear and the experimental design is appropriate to test them.

The approach based on MALDI-TOF analysis to identify snail species is innovative and can improve the reproducibility and accuracy in snail species identifications. The number of samples and quality controls included in this study are well characterized. A total of 530 field specimens belonging to nine species were used. MS spectral analysis allowed the correct identification of all the species applying the statistical tests embedded into the MALDI-TOF Biotyper software. The entire study received ethical approval.

However, from a methodological point of view there some points that need to be addressed as described below. 

Comments:

- A comparison between the number of peaks, intensity, signal to noise and reproducibility (CVs) using the snails' foot and the head should be provided.

- The authors wrote: “The difference between these protocols is not only in terms of the body parts used but also on the extraction solution volume (70% formic acid, 50% acetonitrile) and the homogenisation method (glass powder or tungsten beads)”. Are these two different solutions or the authors used a mixture of 50% acetonitrile and 35% formic acid and 15% water? This should be better clarified.

- Why the authors used this composition of the extraction solution and did not optimized the ratio acetonitrile/water to observe the difference in protein extraction? It is not reported the optimization of extraction solution based on the organic solvent type, concentration and type of acid. 

- The 15, 30 and 40ul of extraction solution were optimized based on the weight of snails’ foots and heads? This point should be clarified.

- The authors should better specify that the different protocols tested had several variables: 1) volume of extraction solution, 2) homogenization beads, 3) snail parts and 4) snail species.

- The authors state: “the choice of the body part that provides greater percentage of protein, and the simplicity of the protocol..”. The “greater percentage of proteins” should be changed in number of MS peaks.

- A saturated solution of alpha-cyano-4-hydroxycinnamic acid dissolved in 50% acetonitrile (v/v), 10% trifluoroacetic acid (v/v) was used. Saturated means 10 mg/mL? For method optimization, did the authors test different matrices such as sinapinic acid dissolved in different concentration of acids? This point should be addressed.

- The MALDI-TOF data should be deposited in a public repository such as PRIDE mass spectrometry data repository since they constitute a useful resource for the community.

Reviewer #2: -Are the objectives of the study clearly articulated with a clear testable hypothesis stated? Yes

-Is the study design appropriate to address the stated objectives? Yes 

-Is the population clearly described and appropriate for the hypothesis being tested? No cf Summary and general comments

-Is the sample size sufficient to ensure adequate power to address the hypothesis being tested? Yes

-Were correct statistical analysis used to support conclusions? Yes

-Are there concerns about ethical or regulatory requirements being met? No

**Results**

-Does the analysis presented match the analysis plan?

-Are the results clearly and completely presented?

-Are the figures (Tables, Images) of sufficient quality for clarity?

Reviewer #1: The analyses performed in this study are functional to test the hypothesis of using MALDI-TOF approach for snail species characterization. The images and tables presented are of good quality and allow the reader to evaluate the results obtained during the different analyses. There are some points on the results part that need to be addressed according to this reviewer. 

Comments:

- In Figure 3, the authors should clearly define the MS parameters to define the best extraction protocol. 

- A direct comparison between MALDI-TOF MS spectra obtained from frozen and ethanol-stored samples should be performed to evaluate the effect of sample storage conditions on the spectra quality.

- The authors should compare the MALDI-TOF MS spectra from Schistosoma-infected snails to the uninfected ones.

Reviewer #2: -Does the analysis presented match the analysis plan? Yes

-Are the results clearly and completely presented? Yes

-Are the figures (Tables, Images) of sufficient quality for clarity? Yes

**Conclusions**

-Are the conclusions supported by the data presented?

-Are the limitations of analysis clearly described?

-Do the authors discuss how these data can be helpful to advance our understanding of the topic under study?

-Is public health relevance addressed?

Reviewer #1: The conclusions are supported by the data provided and the authors describe in the conclusion the potentials of this study and further developments that will be needed to apply it in a clinical settings. This study is relevant for the public health system.

Reviewer #2: -Are the conclusions supported by the data presented? Yes

-Are the limitations of analysis clearly described? Yes

-Do the authors discuss how these data can be helpful to advance our understanding of the topic under study? Yes

-Is public health relevance addressed? Yes

**Editorial and Data Presentation Modifications?**

Reviewer #1: (No Response)

Reviewer #2: Abstract line 28-29: “…has revolutionized clinical microbiology and mycology…”, mycology is a part of clinical microbiology. 

Author summary line 48: “…blood flukes of the genus Schistosoma which are released by freshwater snails…” only the cercarial stage is released. 

Ref 8: is a French classic, but it is now a bit outdated. Recent data on Schistosomiasis epidemiology can be found in the SCORE special issue of Am. J. Trop. Med. Hyg. (N° 103 Suppl 1, 2020). 

line 83: B. umbilicatus 

line 90: I suggest “depend mainly” or “relies mainly on” instead of “often depends”. 

line 98-99: “MALDI-TOF (Matrix Assisted Laser Desorption / Ionization Time-of-Flight) mass spectrometry” could be changed to MALDI-TOF MS (Matrix Assisted Laser Desorption / Ionization Time-of-Flight Mass Spectrometry) as MALDI-TOF MS is the abbreviation used. 

line 108: Stephan et al. (2) were the first to report MALDI-TOF use for specific identification in Mollusca. 

line 118: “The weather is warm all year round.” This sentence is not informative as average, and min/max are given in the next sentence. 

line 151: “extraction solution proportion” instead of “volume”? 

Line 351: It could be interesting to cite the paper by Stephan et al. (2) as it is the first to report MALDI-TOF use for specific identification in Mollusca. 

Line 339: “We performed a qPCR for the detection of S. mansoni and S. haematobium” Interestingly, the probes from Cnops et al. and the Sh-Fw and Sh-Rv primers, could hybridize with other species of the S. haematobium complex, thus S. intercalatum or S. currassoni are detected. I propose to change to “We performed a qPCR for the detection of S. mansoni and S. haematobium species complex”. 

line 259-260: “an unsupervised test” instead of “the unsupervised test” 

line 263: “as confirmed by Flex Analysis”, I suppose by visual inspection? 

line 360: “adequate” instead of enough? 

Table 2: M. tuberculata instead of “Melanoides tuberculate”, or provide full-name for all species. 

Table 3: “Good quality Spectra” instead of “Good Spectra” 

Table 3: For the fifth column, did the percentage correspond to percentage of similarity against best match in Genbank? Please precise in the column title. 

Table 3: Please provide a foot-note explanation of the character “*” (as in Table 2 legend) 

Figure 3 panel B: To which Principal Component correspond x and y axis of the 2D plot? 

Figure 5 panel A: To which Principal Component correspond x and y axis of the 2D plot? The variance explained by each component could also be an interesting data.

**Summary and General Comments**

Reviewer #1: Taken together, the data and the application of MALDI-TOF presented in this manuscript have the potential to advance significantly the malacology field. The number of samples and the deepness of their characterization using a combination of techniques makes this study novel and significant. However, according to this reviewer, there are some points that need to be addressed before acceptance in Plos Neglected Tropical Diseases journal. The specific points are related to the methods and results sections as described before.

Reviewer #2: Title: MALDI-TOF mass spectrometry for the identification of freshwater snails, including intermediate hosts of schistosomes in Senegal 

In this paper, Hamlili et al. propose the MALDI-TOF Mass-Spectrometry as a new approach to identify freshwater snail’s vectors of human schistosomiasis. The authors have optimized a reliable protocol to produce MALDI-TOF spectra using feet of fresh, frozen or ethanol stored snails, and begin the implementation of a spectral database containing the main vectors of schistosomiasis. 

The methodology proposed by the author is innovative and address the important problem of the identification of schistosomiasis vector when lacking malacology expertise. The paper is well-written and the methodology is appropriate. It also highlight the problem of establishing a MALDI-TOF database with species of doubtful taxonomic status. 

I precise that I’m not a malacologist and therefore the very interesting questions raised in this paper regarding molecular and morphologic identification or the taxonomic status of C. bulimoides are outside my field of expertise. 

Main remarks: 

The global design of the study is clear but the number of specimens used in the different groups is hard to follow. Please clarify the relevant parts in materiel and methods, results and table sections. To my opinion the better option is to provide a flowchart of the different experiments. For instance, in table 3 (identification results of ethanol-stored snails) the total of column “Specimen collection” is 210 but the addition of B. pfeifferi, B. truncatus and B. senegalensis specimen is 160. I understand with this table that 80 specimens were tested and 40 used for blind-tested. The use of the 40 remaining specimens is unclear to me. In parallel, the text refers to 244 ethanol-stored snails, minus 13 used for testing protocols. 

line 151: in order to reproduce the protocol, information regarding the amount of snail tissue is important. Was the volume of tissue the same as the mix volume? 

- line 339 The primer combinations used has not been tested with all other larval trematodes that could potentially infest the study snails. Bulinus and Biomphalaria can be infested with Echinostoma or Paramphistomidae (3,4). According to Hamburger et al. (5), Sh-Fw/Sh-Rv don’t hybridize with Echinostoma. Were the ethanol preserved snails visually inspected during dissection to assess the absence or presence of trematodes? The infection with non-detected larval trematodes could introduce an important bias in the acquired spectra. 

-Please precise if the snail MALDI-TOF database is available on request.

Minor remarks: 

line 173: the composition of the commercial Bruker standard solvent (OS solution) for resuspension is: Acetonitrile 50%, trifluoroacetic acid (TFA) 5%, Water 45% (Sigma ref 900666). In ref 34 the composition of OS solution is the same as in the present paper but it cite another reference from Marseille (1) with 2.5% of TFA. What is the purpose of increasing the proportion of TFA? 

- line 305: Did you succeed to obtain sequence for all specimens? 

- Line 368 and line 812 table 1: Disruption using tungsten beads was not tested with Foot and Head-Foot in frozen specimens. What is the reason? 

References: 

1. Fournier P-E, Couderc C, Buffet S, Flaudrops C, Raoult D. Rapid and cost-effective identification of Bartonella species using mass spectrometry. J Med Microbiol. 1 sept 2009;58(9):1154‑9. 

2. Stephan R, Johler S, Oesterle N, Näumann G, Vogel G, Pflüger V. Rapid and reliable species identification of scallops by MALDI-TOF mass spectrometry. Food Control. 1 déc 2014;46:6‑9. 

3. Chappell LH. Freshwater snails of Africa and their medical importance (second edition). Trans R Soc Trop Med Hyg. nov 1994;88(6):717. 

4. Laidemitt MR, Brant SV, Mutuku MW, Mkoji GM, Loker ES. The diverse echinostomes from East Africa: With a focus on species that use Biomphalaria and Bulinus as intermediate hosts. Acta Trop. 1 mai 2019;193:38‑49. 

5. Hamburger J, Abbasi I, Ramzy RM, Jourdane J, Ruppel A. Polymerase chain reaction assay based on a highly repeated sequence of Schistosoma haematobium: a potential tool for monitoring schistosome-infested water. Am J Trop Med Hyg. 2001;65(6):907‑11.

PLOS authors have the option to publish the peer review history of their article (what does this mean?). If published, this will include your full peer review and any attached files.

Reviewer #1: No

Reviewer #2: Yes: Antoine Huguenin
---

## [Decision Letter · Decision Letter 1]

19 Jul 2021

Dear Pr. Parola,

Thank you very much for submitting your manuscript "MALDI-TOF mass spectrometry for the identification of freshwater snails from Senegal, including intermediate hosts of schistosomes." for consideration at PLOS Neglected Tropical Diseases. As with all papers reviewed by the journal, your manuscript was reviewed by members of the editorial board and by several independent reviewers. The reviewers appreciated the attention to an important topic. Based on the reviews, we are likely to accept this manuscript for publication, providing that you modify the manuscript according to the review recommendations. For instance, according to reviewer #1, the authors should deposit the raw data in a public repository. Several corrections/changes in the text and figure 2 are indicated by reviewer #2. These modifications should be done before the manuscript is deemed acceptable for publication. Due to the long time past since the original submission, I am willing to accept the revised manuscript without sending it back to the reviewers, provided all suggested changes are made.

Sincerely,

Igor C. Almeida

Associate Editor

Sergio Oliveira

Deputy Editor

Reviewer's Responses to Questions

**Key Review Criteria Required for Acceptance?**

**Methods**

-Are the objectives of the study clearly articulated with a clear testable hypothesis stated?

-Is the study design appropriate to address the stated objectives?

-Is the population clearly described and appropriate for the hypothesis being tested?

-Is the sample size sufficient to ensure adequate power to address the hypothesis being tested?

-Were correct statistical analysis used to support conclusions?

-Are there concerns about ethical or regulatory requirements being met?

Reviewer #1: The authors have incorporated the suggestions of this reviewer improving the methodological details.

Reviewer #2: Yes to all questions

**Results**

-Does the analysis presented match the analysis plan?

-Are the results clearly and completely presented?

-Are the figures (Tables, Images) of sufficient quality for clarity?

Reviewer #1: The authors have replied to the majority of the comments raised by this reviewer.

- However, the authors should deposit all the MALDI-MS data in a public repository such as ProteomeXchange, Massive and others to make available for the entire research community. This reviewer considers extremely important this point since the raw data should be publicly shared with the research community.

- Please state in the legend of Figure 3 the protocol that was chosen to build the reference database and perform the analysis.

Reviewer #2: Yes to all questions

**Conclusions**

-Are the conclusions supported by the data presented?

-Are the limitations of analysis clearly described?

-Do the authors discuss how these data can be helpful to advance our understanding of the topic under study?

-Is public health relevance addressed?

Reviewer #1: The revised version of the manuscript has substiantially improved the conclusions.

Reviewer #2: Yes to all questions

**Editorial and Data Presentation Modifications?**

Reviewer #1: This reviewer believes that the authors should deposit all the MALDI-MS data in a public repository such as ProteomeXchange, Massive and others to make available for the entire research community. This part is essential to further develop the method reported in this manuscript.

Reviewer #2: See general comments

**Summary and General Comments**

Reviewer #1: The revised version of the manuscript has been substantially improved following the suggestions of the reviewers.

However, there are some points that need to be addressed before publication in Plos Neglected Tropical Diseases journal.

In particular, the authors should make available the raw data for the entire research community and deposit these data in a public repository.

Due to that, according to this reviewer, the manuscript can be accepted after minor revisions.

Reviewer #2: I find the author’s responses to the reviewers' comments quite satisfactory. The revised manuscript seems clearer to me. 

I have a few remarks/suggestions left:

- Line 38 : log-score instead of log score

- Line 76 : the sentence «…, whereas S. mekongi, S. guineensis and S. intercalatum have a lower global prevalence » is not clear as these species have a restricted geographic distribution (South-East asia for S. mekongi, Cameroon, Equatorial Guinea, Gabon, Nigeria, and Sao Tomé for S. guineensis). I am not sure that it is necessary to mention these species in the manuscript

- Line 274-276: “The visual comparison of MS profiles using an unsupervised statistical test (PCA, ClinProtools software) revealed that the dots corresponding to MS spectra from H2, FH, F are separated from those of H1, H3 (Fig 3B).” instead of “The visual comparison of MS profiles using an unsupervised statistical test (PCA, ClinProtools software) revealed a separation of the dots corresponding to MS spectra from H2, FH, F and H1, H3 (Fig 3B).”

- Line 289: 127 of the 186 frozen snail instead of “127 of the 186 snail ».

- Line 399 : « …as Stephan et al. were the first… » or « …as the paper by Stephan et al. was the first… » instead of « …as Stephan et al. was the first… » 

- Line 514 : « it could be used to circumvent molecular limitations. ». It is not clear for me which molecular limitation are circumvent. Are these the limitations related to the price and the time required for molecular biology?

- Line 556 : organic preservation solution instead of « organic buffer » as ethanol has no buffering effect. 

- Line 586: “…efficient to discriminate two cryptic species that are hard to be distinguished morphologically… » If I understand correctly the specimens used for the blind test came from the second field collection and were not identified by molecular biology. It is therefore not possible to be sure if MALDI-TOF is able to discriminate morphologically close species such as B. senegalensis and B. forskalii.

- Line 1012 – Figure 5: Panel D, mentioned in legend of figure 5, is missing in the file Figure5.tiff 

- Figure 2: In my opinion, adding this figure helped clarify the workflow. Perhaps, if that does not overload the figure, it would be interesting to mention, in boxes B and C, the species respectively used for the creation of the base and for the blind test. Or alternatively, refer to the tables concerned.

PLOS authors have the option to publish the peer review history of their article (what does this mean?). If published, this will include your full peer review and any attached files.

Reviewer #1: No

Reviewer #2: Yes: Antoine Huguenin

Figure Files:

Data Requirements:

Reproducibility:

References

---

## [Editor Report · Decision Letter 2]

12 Aug 2021

Dear Pr. Parola,

We are pleased to inform you that your manuscript 'MALDI-TOF mass spectrometry for the identification of freshwater snails from Senegal, including intermediate hosts of schistosomes.' has been provisionally accepted for publication in PLOS Neglected Tropical Diseases.

Best regards,

Igor C. Almeida

Associate Editor

Sergio Oliveira

Deputy Editor

---

## [Editor Report · Acceptance letter]

7 Sep 2021

Dear Pr. Parola,

We are delighted to inform you that your manuscript, "MALDI-TOF mass spectrometry for the identification of freshwater snails from Senegal, including intermediate hosts of schistosomes," has been formally accepted for publication in PLOS Neglected Tropical Diseases.

Best regards,

Shaden Kamhawi

co-Editor-in-Chief

Paul Brindley

co-Editor-in-Chief
